# Moving toward Precision Medicine in Acute Coronary Syndromes: A Multimodal Assessment of Non-Culprit Lesions

**DOI:** 10.3390/jcm12134550

**Published:** 2023-07-07

**Authors:** Michele Bellino, Angelo Silverio, Luca Esposito, Francesco Paolo Cancro, Germano Junior Ferruzzi, Marco Di Maio, Antonella Rispoli, Maria Giovanna Vassallo, Francesca Maria Di Muro, Gennaro Galasso, Giuseppe De Luca

**Affiliations:** 1Department of Medicine, Surgery and Dentistry, University of Salerno, 84084 Baronissi, Italy; michelebellino8@gmail.com (M.B.); asilverio@unisa.it (A.S.); lucesposito@unisa.it (L.E.); fcancro@unisa.it (F.P.C.); germano.ferruzzi.jr@gmail.com (G.J.F.); mdimaio@unisa.it (M.D.M.); antonellarispoli@gmail.com (A.R.); mvassallo@unisa.it (M.G.V.); 2Structural Interventional Cardiology, Department of Clinical and Experimental Medicine, Clinica Medica, Careggi University Hospital, 50139 Florence, Italy; fdimuro94@gmail.com; 3Division of Cardiology, AOU “Policlinico G. Martino”, Department of Clinical and Experimental Medicine, University of Messina, 98166 Messina, Italy; gdeluca@uniss.it; 4Division of Cardiology, IRCCS Hospital Galeazzi-Sant’Ambrogio, 20161 Milan, Italy

**Keywords:** acute coronary syndrome, multivessel disease, non-culprit lesion, intracoronary imaging, IVUS, OCT, coronary computed tomography angiography, cardiac magnetic resonance

## Abstract

Patients with acute coronary syndrome and multivessel disease experience several recurrent adverse events that lead to poor outcomes. Given the complexity of treating these patients, and the extremely high risk of long-term adverse events, the assessment of non-culprit lesions becomes crucial. Recently, two trials have shown a possible clinical benefit into treat non-culprit lesions using a fraction flow reserve (FFR)-guided approach, compared to culprit-lesion-only PCI. However, the most recent FLOW Evaluation to Guide Revascularization in Multivessel ST-elevation Myocardial Infarction (FLOWER-MI) trial did not show a benefit of the use of FFR-guided PCI compared to an angiography-guided approach. Otherwise, intracoronary imaging using optical coherence tomography (OCT), intravascular ultrasound (IVUS), or near-infrared spectroscopy (NIRS) could provide both quantitative and qualitative assessments of non-culprit lesions. Different studies have shown how the characterization of coronary lesions with intracoronary imaging could lead to clinical benefits in these peculiar group of patients. Moreover, non-invasive evaluations of NCLs have begun to take ground in this context, but more insights through adequately powered and designed studies are needed. The aim of this review is to outline the available techniques, both invasive and non-invasive, for the assessment of multivessel disease in patients with STEMI, and to provide a systematic guidance on the assessment and approach to these patients.

## 1. Introduction

Approximately half of patients suffering from ST-elevation myocardial infarction (STEMI) present with multivessel coronary artery disease (CAD) [1]. Patients with STEMI and multivessel disease (MVD) experience several recurrent adverse events that lead to poor outcomes [2,3,4]. According to the 2017 European Society of Cardiology (ESC) guidelines, routine revascularization of non-culprit lesions in STEMI patients with MVD should be considered before hospital discharge (class II, level of evidence A) [5], while in 2021, in the American College of Cardiology/American Heart Association/Society for Cardiovascular Angiography & Interventions (ACC/AHA/SCAI) Guidelines for Coronary Artery Revascularization, percutaneous coronary intervention (PCI) of non-culprit lesions, planned as staged procedure after successful primary PCI, is recommended in patients with STEMI and MVD who are hemodynamically stable (class I, level of evidence A) [6].

Given the complexity of treating these patients, and the extremely high risk of long-term adverse events, the assessment of non-culprit lesions becomes crucial. Recently, two trials have shown a possible clinical benefit into treat non-culprit lesions using a fraction flow reserve (FFR)-guided approach compared to a culprit-lesion-only PCI [7,8]. However, the most recent FLOW Evaluation to Guide Revascularization in Multivessel ST-elevation Myocardial Infarction (FLOWER-MI) trial did not show a benefit to the use of FFR-guided PCI compared to an angiography-guided approach [9]. Intracoronary imaging using optical coherence tomography (OCT), intravascular ultrasound (IVUS) or near-infrared spectroscopy (NIRS) could provide both quantitative and qualitative assessment of non-culprit lesions. Different studies have shown how the characterization of coronary lesions with intracoronary imaging could lead to clinical benefits in these peculiar groups of patients [10,11,12].

The aim of this review is to outline the available techniques, both invasive and non-invasive, for the assessment of MVD in patients with STEMI, and to provide a systematic guidance on the assessment and approach to these patients.

## 2. Risk of Non-Culprit-Related Events in Patients with ACS and Evidence from Literature Regarding Management

Several studies have evaluated the long-term risk of non-culprit-lesion-related events. The Preventive Angioplasty in Myocardial Infarction Trial (PRAMI) evaluated the efficacy of immediate multivessel PCI versus culprit-lesion-only PCI in 465 patients with STEMI and MVD. The patients were randomized into either immediate multivessel PCI or culprit-lesion-only PCI, and it was found that immediate multivessel PCI was associated with a significant reduction in the risk of the composite endpoint of cardiac death, non-fatal myocardial infarction, or refractory angina at three years follow-up (HR 0.35, 95% CI 0.21–0.58, *p* = 0.001), with an absolute risk reduction of 14 percentage points in the preventive-PCI group [13]. Moreover, the Complete versus Lesion-only Primary PCI (CvLPRIT) trial compared early multivessel PCI (at index procedure or staged at index admission) versus culprit-lesion-only PCI in 296 patients with STEMI and MVD. This study showed that immediate multivessel PCI was associated with a significant reduction in the risk of the composite endpoint of all-cause mortality, recurrent myocardial infarction, heart failure, and ischemia-driven revascularization at 12 months follow-up (10.0% vs. 21.2%, HR 0.45, 95% CI 0.24–0.84, *p* = 0.009) [14]. In addition, The Danish Study of Optimal Acute Treatment of Patients With ST-Elevation Myocardial Infarction—Primary PCI in Patients With ST-elevation Myocardial Infarction and Multivessel Disease: Treatment of Culprit Lesion Only or Complete Revascularization (DANAMI-3 PRIMULTI) trial also evaluated the efficacy of early multivessel PCI (staged at index admission) versus culprit lesion PCI in 627 patients with STEMI and MVD, and found that immediate multivessel PCI was associated with a significant reduction in the risk of major adverse cardiac events (MACE) compared to culprit lesion PCI at one year follow-up (HR 0.56, 95% CI 0.38–0.83, *p* = 0.004) [7]. More recently, the FFR Guided Revascularization Versus Conventional Strategy in Acute STEMI Patients With MVD (COMPARE-ACUTE) trial evaluated in 885 patients with STEMI and MVD; the effectiveness of culprit-lesion-only treatment versus complete revascularization and demonstrated a significant reduction of the composite primary endpoint (all-cause mortality, non-fatal MI, any revascularization, and cerebrovascular events) in patients who underwent complete revascularization (HR 0.35, 95% CI 0.22–0.55, *p* < 0.001) [8]. However, the aforementioned trials were not powered enough to detect significant differences between the components of the composite primary endpoint, when taken individually, and the composite primary endpoints were mainly powered by the ischemia-driven revascularization, with a non-significant difference in the recurrence of MI and all-cause mortality between the two groups.

Finally, the recent Complete vs. Culprit-only Revascularization to Treat Multi-vessel Disease After Early PCI for STEMI (COMPLETE) trial was the largest trial evaluating the culprit-lesion-only PCI vs. complete revascularization, enrolling 4041 patients with STEMI and MVD disease, and found that complete revascularization was associated with a significant reduction in the risk of the composite endpoint of cardiovascular death, recurrent MI, or ischemia-driven revascularization at three years follow-up (HR 0.51, 95% CI 0.43–0.61, *p* < 0.001) [15]. This was the first study to demonstrate that routine non-culprit lesion PCI with the goal of complete revascularization confers a reduction in the long-term risk of cardiovascular death or recurrent MI, regardless of the timing of the intervention (immediate or deferred) [15]. Notably, in this trial, although the FFR was used to evaluate non-culprit lesions that had an angiographically determined degree of stenosis between 50 and 69% the percentage of patients undergoing intracoronary physiology was extremely low. The poor use of this technique has raised several questions regarding the number of PCIs that could have been avoided [16]. However, the recent FLOWER-MI study showed no significant superiority of using an FFR-guided approach over angiography-guided revascularization in patients with STEMI and MVD [9]. On the other hand, non-significant lesions at intracoronary physiology may hide unstable plaques that could precipitate further events; this could highlight the possibility of an imaging-guided complete revascularization with a more complete qualitative assessment of the plaque.

Recently, a metanalysis including all the aforementioned trials, including a total of 6528 patients, demonstrated that complete revascularization based on PCI of non-culprit lesions decreases cardiovascular mortality and the recurrence of MI and repeated revascularization [17].

The most recent Direct Complete Versus Staged Complete Revascularization in Patients Presenting With Acute Coronary Syndromes and Multivessel Disease (BIOVASC) trial evaluated the timing of complete revascularization in 764 ACS patients, randomized to immediate multivessel PCI at index procedure or staged multivessel PCI within 6 weeks, showing that immediate revascularization at the index procedure is non-inferior to staged revascularization for the primary composite outcome (all-cause mortality, MI, any unplanned ischemia-driven revascularization, or cerebrovascular events); however, immediate complete PCI was associated with a significant reduction in recurrent MI and unplanned ischemia-driven revascularization [18].

Table 1 summarizes the most relevant clinical trial comparing culprit-only vs. complete revascularization in patients with STEMI (Figure 1).

The mechanisms underlying non-culprit-lesion-related events in patients with STEMI are complex and multifactorial. These patients frequently show an extensive disease burden of coronary vessels that could lead to plaque instability and thrombosis via different pathways [10,19]. The inflammatory mediators released during the acute phase of STEMI could promote destabilization of non-culprit plaques, promoting thrombosis or causing plaque rupture and subsequent events [20]. In addition, the inflammatory trigger may promote endothelial damage in the microcirculation, leading to microvascular dysfunction, which may facilitate ischemia in the myocardial areas supplied by non-culprit vessels [21]. Moreover, non-culprit lesions may eventually undergo disease progression with an increase in plaque size and possible plaque destabilization, also considering the higher prevalence in these patients of cardiovascular risk factors such as diabetes, dyslipidemia, hypertension, and chronic kidney disease, all of which are major contributors to the disease progression of non-culprit lesions [22,23].

Understanding the pathophysiological mechanisms underlying the risk of future events related to the presence of non-culprit lesions emphasizes the importance of an accurate assessment of multivessel disease in patients with STEMI in order to choose the best strategy for the treatment of these patients and improve their outcomes [24].

## 3. Invasive Assessment

### 3.1. Searching for Ischemia: Can We Rely on Coronary Physiology?

While randomized clinical trials have demonstrated a significant reduction in adverse events in MI patients undergoing complete versus culprit-only revascularization [25], it is not clear which is the optimal strategy to assess the severity of non-culprit lesions (NCLs). Coronary angiography is the most common and quickly available tool for the evaluation of NCLs during the index procedure. Moreover, the trials that demonstrated a prognostic benefit of complete revascularization used an angiography-guided strategy in most cases [7,8,13,14,15]. However, coronary angiography has several limitations. First, there is a significant discrepancy between angiographical assessment and functional significance [26,27]. This discordancy may be even more evident in patients presenting with ACS, in whom the degree of NCLs may be overestimated by approximately 10% in the acute setting, primarily due to a more pronounced vasoconstriction [28]. Indeed, the treatment of all angiographically significant lesions may lead to unnecessary PCI procedures. To overcome these limitations, a physiology-guided strategy with fractional flow reserve (FFR) or non-hyperemic pressure ratios (NHPRs) is a reasonable option for the assessment of NCLs. In patients with chronic coronary syndrome (CCS), physiology-guided PCI has shown to be superior to an angiography-guided strategy, and to reduce the risk of urgent revascularization compared to medical therapy alone [29,30]. However, these findings cannot be automatically translated into the management of NCLs in patients with ACS. Compared to patients with CCS, ACS patients have a different biological and clinical profile, that is reflected by the high risk of recurrent events [31]. In these patents, the NCL-related risk of adverse events may be closely related to plaque morphology rather than to functional significance. Therefore, the deferral of non-flow limiting stenoses may still expose to the risk of adverse events related to the presence of high-risk morphological features, such as a thin fibrous cap, high lipid burden, and inflammation. Moreover, transient changes in microvascular physiology, in the acute phase of an MI, may affect the reliability of both hyperemic and non-hyperemic indexes [32,33,34]. These changes occur not only in the territory of the infarct-related artery, but also in areas of myocardium supplied by NCLs, especially in case of large infarcts [35]. FFR is the whole cycle ratio between distal coronary pressure (Pd) and aortic pressure (Pa) in a condition of maximal hyperemia, usually induced by the administration of adenosine. However, in the acute or subacute phase of an MI, hyperemic flow may be reduced, and tends to normalize within months from the acute event [36]. Several alterations in microvascular function are responsible for the reduction of hyperemic flow, such as a reduced response to adenosine, the enhanced microvascular vasoconstriction, and the microvascular compression due to edema, and increased end-diastolic pressures [37]. Because the reliability of FFR measurements is related to the induction of the maximal hyperemic state, a reduced hyperemic flow may induce the underestimation of the functional significance of NCLs [38,39]. Furthermore, several real-world data showed that FFR is significantly underused by the interventional cardiology community in the assessment of angiographically intermediate stenoses, even in patients with stable CAD [40,41]. Considering the acute clinical setting, the additional procedural time, and the need for adenosine administration, FFR might be even less adopted in patients with ACS.

Moreover, additional questions about the role of FFR in patients with MVD were raised by the FAME 3 study. In this trial, an FFR-guided PCI strategy did not meet the noninferiority margin compared to coronary artery bypass grafting in patients with three-vessel disease [42,43]. However, although about 40% of the patients enrolled had a non-ST segment elevation ACS, this trial was not specifically designed to assess the role of FFR in patients with ACS and MVD, and STEMI was an exclusion criteria.

On the other hand, NHPRs measure resting Pd/Pa during the entire cardiac cycle, or during specific phases of the diastole, depending on the specific index [39]. However, during the acute phase of an MI, resting flow may be increased, probably due to the compensatory hyperkinesia in non-infarct related territories [35,37]. Therefore, NHPRs may overestimate the functional significance of NCLs.

To date, two randomized clinical trials compared physiology-guided vs. angiography-guided PCIs of NCLs in patients with MI. The FLOWER-MI trial randomized 1171 patients with STEMI and MVD to either the FFR- or angiography-guided complete revascularization strategy. The number of NCL-PCI was significantly lower in the FFR-guided group (66% vs. 97%). At 12 months, there was no difference between the two groups for the primary composite outcome of death, MI, or urgent revascularization (5.5% FFR-guided vs. 4.2% angiography-guided, hazard ratio (HR) 1.32; 95% confidence interval 0.78–2.23; *p*= 0.31). The Kaplan–Meier curves appeared to separate late, in support of an angiography-guided strategy. However, due to the very low event rate and the wide confidence interval, the authors state that no definite conclusions can be drawn based on these data [9]. In the FRAME-MI trial, 562 patients with MI (both STEMI and NSTEMI) were randomized to FFR- or angiography-guided complete revascularization. The original sample size was 1292 patients, but the trial was stopped prematurely by the executive committee due to slow recruitment because of the COVID-19 pandemic. NCL-PCI was performed less frequently in the FFR-guided group (64% vs. 97%). At a median follow-up of 3.5 years, the primary endpoint of death, MI, or repeat revascularization was reduced in the FFR-guided arm compared to the angiography-guided arm (7.4% vs. 19.7%, HR 0.43, *p* = 0.003) [44]. However, due to the relatively small sample size and the low event rate in both trials, these contradictory results should be interpreted with caution, and larger-scale randomized clinical trials are needed to clarify the role of physiology-guided complete revascularization in patients with MI.

### 3.2. Intravascular Imaging: The Dilemma of the “Vulnerable Plaques”

Over the past decades, several studies showed that sudden cardiac death and MI usually arise from the rupture, erosion, or ulceration of high-risk atherosclerotic plaques, followed by the thrombotic occlusion of a coronary artery. This observation led to the identification of the so-called “vulnerable” plaque, defined as any plaque at risk of progression and instability, with the potential of becoming a culprit lesion of an ACS. These plaques are usually characterized by specific high-risk features, like the presence of a large lipidic burden, a necrotic core, and a thin fibrous cap (tin-cap fibroatheroma, TFCA) [45]. While the association between vulnerable plaques and adverse coronary events was first observed in autopsy findings, the introduction of intravascular imaging (IVI) allowed an in vivo assessment of the natural history of coronary atherosclerosis and of the mechanisms of ACS [46]. The identification of high-risk atherosclerotic plaques is particularly relevant in the assessment of NCLs in patients with MI, given the high risk of recurrent events in this setting [4,31,32,47]. In the PROSPECT study, 697 patients with a recent ACS, treated with PCI of all culprit lesions, underwent three-vessel angiography and intravascular ultrasound (IVUS) for the assessment of non-flow limiting plaques. At a mediant follow-up of 3.4 years, the cumulative incidence of major adverse cardiovascular events (MACE) was 20.4%, nearly equally distributed between culprit- (12.9%) and non-culprit-related (11.6%) events. Of note, multivariable analysis showed that a plaque burden ≥ 70%, minimal lumen area (MLA) ≤ 4 mm^2^, and the presence of a TCFA were independently associated with the incidence of adverse events [10]. However, due to its limited resolution, IVUS could not provide sufficient data on specific features of plaque vulnerability, including lipid burden, thrombus formation, and fibrous cap thickness. To partially overcome these limitations, near-infrared spectroscopy (NIRS) was combined with IVUS for the assessment of vulnerable plaques. Indeed, this technology combines the ability of NIRS to evaluate lipid content, and the information provided by IVUS regarding plaque burden and MLA. In the PROSPECT-II study, NIRS-IVUS was used for the evaluation of NCLs in patients with a recent MI. The authors found that most of adverse events at follow-up were related to mild lesions that were non-flow limiting at the time of the index event, and that large plaque burden and high lipidic content were independent predictors of NCL-related adverse events [48].

Optical coherence tomography (OCT) is a near-infrared, light-based imaging technology, with an axial resolution of approximately 10–15 μm, that generates high-resolution reconstructions of the vessel microstructure [49]. Given its ability to evaluate plaque composition, OCT has become the gold standard for the assessment of plaque vulnerability (Table 2). Aside from the typical features of high-risk plaques (i.e., lipid pool and fibrous cap thickness), OCT allowed the detection of additional aspects of vulnerable plaques, like macrophages infiltration and neovascularization (Figure 2 and Figure 3) [50,51]. Several OCT studies assessed the association between vulnerable plaque features and clinical outcomes. In the prospective CLIMA study, Prati et al. found that the presence of MLA < 3.5 mm^2^, a lipid arc with circumferential extension > 180°, and macrophages, were all independently associated with the risk of cardiac death and target segment MI [52]. Kubo et al., in a prospective study on 1378 patients and 3533 NCLs, showed that, at a median follow-up of 6 years, NCLs that were both lipid-rich and TCFA were associated with approximatively a 17-fold increase in the risk of subsequent ACS [11]. In the COMBINE-OCT trial, the presence of a TCFA was associated with a nearly five-fold increase in the risk of MACE in diabetic patients with negative FFR [53]. These data show that IVI allows the identification of high-risk plaques with the greatest probability to progress and cause an ACS. However, the optimal treatment of vulnerable plaques remains an unanswered question. While the revascularization of angiographically and/or functionally significant NCLs has shown to reduce adverse events in patients with ACS [25], whether the prophylactic revascularization of non-flow-limiting vulnerable plaques may improve long-term outcome is unknown. In the PROSPECT ABSORB trial, NIRS-IVUS-guided PCI with bioresorbable vascular scaffold (BVS) of non-obstructive lesions with large plaque burden was safe, and was associated with comparable clinical outcomes and significantly higher MLA compared to medical therapy alone at 2 years [54]. However, although attractive, these data should be considered as hypothesis-generating, and should be confirmed in adequately powered randomized clinical trials. To date, there is no evidence supporting routine PCI for angiographically mild high-risk plaques, and future studies are needed to address the unsolved dilemma of the vulnerable plaques.

Medical therapy, especially with novel aggressive lipid lowering agents, is of paramount importance to achieve plaque stabilization. The PACMAN-AMI and the HUYGENS trials assessed the effect of alirocumab and evolocumab, respectively, in addition to high-intensity statin therapy on non-flow-limiting NCLs of patients with a recent ACS. Plaque morphology was assessed by OCT alone in HUYGENS, and with both OCT and NIRS-IVUS in PACMAN-AMI. These studies demonstrated a trend toward plaque stabilization in patients treated with PCSK9-inhibitors compared to patients treated with high-intensity statins alone, with a significant reduction of plaque burden at IVUS and of maximum lipid core burden index at NIRS, and a significant increase in fibrous cap thickness at OCT [55,56]. Further studies should evaluate whether these morphological features of plaque stabilization may translate into improved clinical outcomes.

## 4. Non-Invasive Assessment: Anatomical vs. Functional Tests

The presence of non-culprit lesions which lead to residual ischemia could also be assessed by non-invasive diagnostic techniques. According to latest guidelines, the timing and appropriate non-invasive assessment, including anatomical and functional tests, to detect ischemia and myocardial viability in patients with non-culprit lesions and recent acute coronary syndrome, remains to be determined, and depends on local availability and expertise [5,57].

In recent years, the coronary computed tomography angiography (CCTA) has evolved as a logistically accurate, low-risk, non-invasive test to diagnose or rule-out CCS [58]. In addition, recent advances in CCTA and computational fluid dynamics technologies allow simultaneous acquisition of anatomical parameters, identification of high-risk plaque features, and assessment of non-invasive hemodynamic [59,60], which allow performance of rapid clinical decision making in low-risk patients presenting with acute chest pain in the emergency department [61,62], and to rule-out clinically significant coronary artery disease in low-risk patients with non-ST-segment elevation acute coronary syndrome (NSTE-ACS) [63].

In the setting of acute coronary syndrome and multivessel disease, the vulnerability of non-culprit lesions is an important factor that could predispose to recurrent ischemia; this explains the necessity to identify high-risk plaque characteristics [64]. In this context, the emerging role of high-risk plaque features assessment by CCTA, such as low-attenuation plaque, napkin ring sign, positive remodeling and spotty plaque calcification, provides an opportunity to personalize risk assessment of future acute coronary syndrome events [65,66,67,68]. Several studies were performed to detect the predictive value of high-risk plaque features using CCTA. The PROMISE (Prospective Multicenter Imaging Study for Evaluation of Chest Pain) and SCOT-HEART (Scottish Computed Tomography of the Heart) studies showed an increased number of major cardiovascular events in patients with high-risk plaque features detection by CCTA; however, these results are limited by modest positive predictive value of high-risk plaque features for the identifying of subsequent coronary events [69,70].

Besides the estimation of plaques features, an accurate assessment of hemodynamic parameters is critical for the identification of high-risk untreated non-culprit lesions [71,72]. In recent years, CCTA-derived FFR (FFR-CCTA) has used an advanced fluid dynamic analysis method that combines the advantages of non-invasive CCTA and traditional invasive FFR. This processing technology derives hemodynamic parameters from CCTA image data, in order to quantify the hemodynamic impact of coronary artery stenosis [72,73,74]. In a previous study, Lee and colleagues enrolled patients with documented acute coronary syndrome, and evaluated non-culprit stenosis by CCTA; the authors showed that non-invasive hemodynamic assessment by computational fluid dynamics might help to identify high-risk plaques that subsequently cause the recurrence of acute coronary syndrome [71]. However, the diagnostic accuracy of non-invasive hemodynamic parameters assessed by CCTA for the detection of ischemia in non-culprit vessels is modest, and limited compared to traditional invasive wire-based hemodynamic assessment in patients with recent acute coronary syndrome and multivessel disease [75]. Indeed, Gaur and colleagues investigated the diagnostic performance of non-invasive hemodynamic parameters assessed by CCTA, compared to traditional invasive coronary angiography with FFR, in 124 non-culprit lesions of 60 patients with recent ST-segment elevation myocardial infarction (STEMI); the authors reported that the diagnostic performance of the non-invasive approach for staged detection of ischemia in STEMI patients with multivessel disease is only moderate when compared to invasive FFR [75].

Coronary flow velocity reserve (CFVR) with transthoracic Doppler echocardiography is another non-invasive functional test, which has shown that deferred revascularization is safe and associated with excellent long-term clinical outcomes in patients with untreated non-culprit lesions and recent primary percutaneous coronary intervention (PCI), if CFVR > 2 [76].

In later years, the focus has been placed on non-invasive cardiac magnetic resonance (CMR) imaging that allows simultaneous assessment of left ventricular function, transmural extent of myocardial scar tissue, and stress myocardial perfusion [77,78]. More recently, Everaars and colleagues demonstrated that stress perfusion CMR with adenosine might help to identify residual ischemia in non-culprit territory lesions after recent STEMI, although agreement with invasive FFR assessment at 1 month was moderate [79].

Few and limited studies have investigated residual ischemia in patients with acute coronary syndrome and multivessel disease by non-invasive testing, so further research is needed to clarify whether this approach can be helpful to clinicians in identifying patients with untreated non-culprit lesions with vulnerable features and higher risk of future cardiovascular events.

## 5. Future Perspectives

Currently, the decision of optimal timing to identify the hemodynamic relevance of non-culprit lesions remains controversial in patients with acute coronary syndrome and multivessel disease [33,34,38,80].

The FULL-REVASC study (NCT02862) will compare a PCI strategy for non-culprit lesions guided by immediate versus staged FFR in patients with STEMI and multivessel disease to detect differences in all-cause mortality, non-fatal myocardial infarction, and unplanned revascularization at 1 year [81,82].

The iMODERN trial (NCT03298659) will compare an iFR-guided approach of non-culprit lesions during the acute setting with a deferred stress perfusion CMR-guided strategy during the outpatient follow-up to determine the optimal therapeutic approach for STEMI patients with multivessel disease.

Finally, intracoronary imaging can help identify high-risk features of vulnerable non-culprit lesions; however, the positive predictive value derived from intracoronary imaging for the occurrence of future cardiovascular events is limited [10,52].

The PREVENT study (NCT02316886) will investigate whether preventive PCI of non-significant hemodynamic parameters assessment by invasive FFR in non-culprit stenosis (FFR > 0.80) with vulnerable plaque features and optimal medical therapy may improve the clinical outcome compared with optimal medical therapy alone.

Despite the new light shed by several recent studies on the management of non-culprit lesions in patients with acute coronary syndrome and multivessel disease, several knowledge gaps remain; only large, well-designed and adequately powered randomized trials will be able to answer them.

## 6. Conclusions

Acute coronary syndromes make up a large percentage of the burden of modern-day cardiovascular disease. Although, over the years, we have achieved much progress in identifying the correct timing of intervention and increasingly precise peri-procedural and long-term therapy, further efforts need to be perpetuated to refine the indications of the latest technologies, invasive and non-invasive, in the diagnostic–therapeutic pathway of patients with ACS and MVD (Figure 4).

Within this scenario, the clinical condition takes the lead regarding the timing.

Complete revascularization could be performed at the time of primary PCI in patients with low clinical risk, i.e., low risk NCLs.

On the other hand, complete revascularization in high-risk patients (low ejection fraction, high probability of acute kidney injury, etc.) with high-risk lesions (left main, bifurcation, heavily calcified lesions), should be performed at a staged time after a discussion by the heart team about the exclusion of heart surgery and optimal medical therapy.

Regarding functional tests, they should preferably be avoided in the acute phase, and should be performed at a staged time once better hemodynamic stabilization is achieved, to identify NCLs worthy of undergoing angioplasty.

Instead, invasive coronary imaging, especially OCT, could be adopted to identify culprit lesions in uncertain angiographic pictures and should be adopted, at a staged time, for the proper identification of vulnerable lesions and optimization of angioplasty.

Non-invasive evaluation of NCLs has begun to take ground in this context, but more insights through adequately powered and designed studies are needed.

In the end, the forthcoming advent of new platforms, capable of simultaneously providing anatomical and functional information, will provide more data and enable tailoring the indications for each method and identify the best operational strategy regarding the treatment of NCLs in ACS.

## Figures and Tables

**Figure 1 jcm-12-04550-f001:**
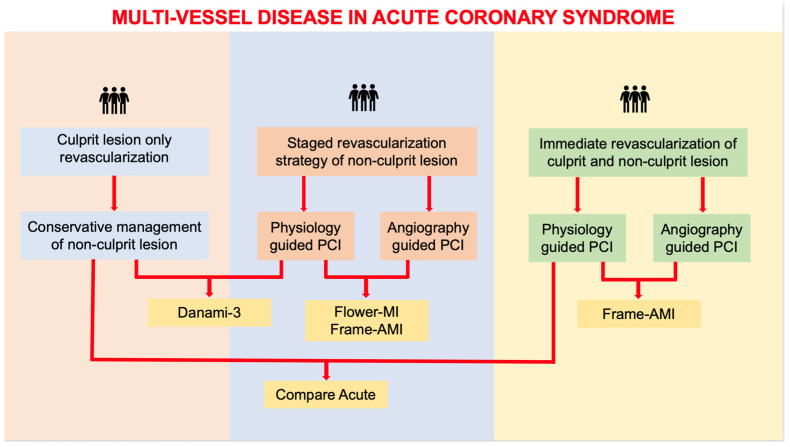
Focuses on the management provided by the main studies regarding management of multi-vessel disease in ACS.

**Figure 2 jcm-12-04550-f002:**
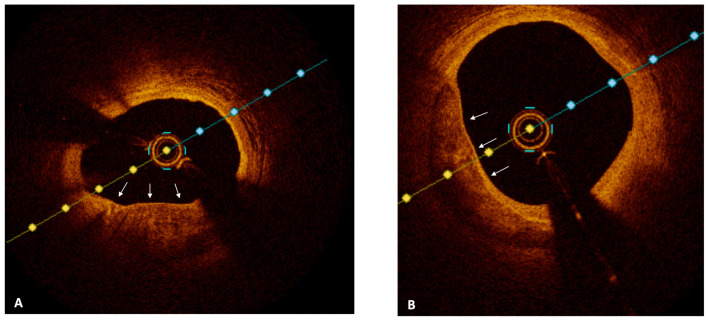
Panel (**A**,**B**); OCT images showing non-culprit lesion with plaque vulnerability characteristics: thin cap fibroatheroma, macrophage present in the fibrous cap, and high lipid core burden (white arrows).

**Figure 3 jcm-12-04550-f003:**
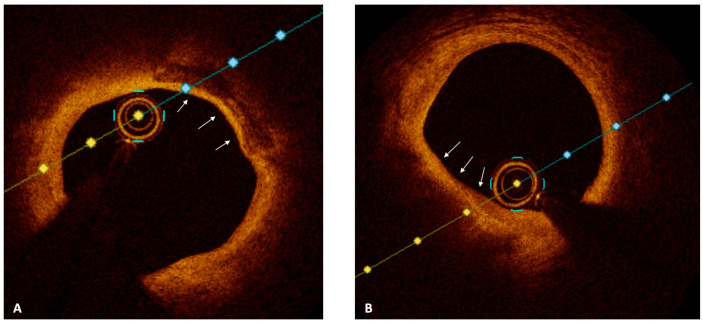
Panel (**A**,**B**); OCT images showing non-culprit lesion with necrotic lipid core (white arrows).

**Figure 4 jcm-12-04550-f004:**
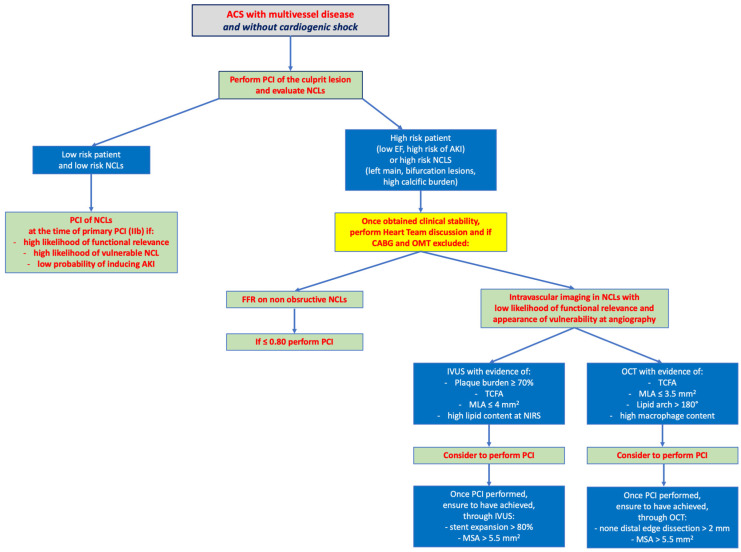
Operational algorithm for the management of multivessel disease in patients with acute coronary syndrome without cardiogenic shock. ACS, acute coronary syndrome; AKI, acute kidney injury; CABG, coronary artery bypass grafting; EF, ejection fraction; FFR, fractional flow reserve; MLA, minimal lumen area; MSA, minimal stent area; NCLs, non-culprit lesions; NIRS, near infrared spectroscopy; OCT, optical coherence tomography; OMT, optimal medical therapy; PCI, percutaneous coronary intervention; TCFA, thin cap fibro-atheroma.

**Table 1 jcm-12-04550-t001:** Clinical trials comparing culprit-only vs. complete revascularization in STEMI.

Study, Year	Registration Number	Population	Timing of Non-Culprit Lesion PCI	Primary Outcomes	Findings
PRAMI, 2013	ISRCTN73028481	234 pts undergoing CR vs. 231 pts receiving C-OR *n* = 465	Index procedure	MACE: CV death, non-fatal MI, refractory angina at 23 months FU.	Preventive MV PCI NCL reduced the risk of adverse CV events (9% vs. 23%, HR 0.35, 95% CI 0.21–0.58, *p* = 0.001)
CvULPRIT, 2015	ISRCTN70913605	150 pts undergoing CR vs. 146 pts receiving C-OR *n* = 296	Index procedure or index admission (staged)	MACE: Death, MI, any repeat revascularization, HF at 12 months FU.	MV PCI reduced the rate of adverse CV events (10.0% vs. 21.2%, HR 0.45, 95% CI 0.24–0.84, *p* = 0.009)
DANAMI-3 PRIMULTI, 2015	NCT01960933	314 pts undergoing CR vs. 313 pts receiving C-OR *n* = 627	Index admission (staged)	MACE: Death, re-infarction, IDR at 27 months FU.	MV FFR-guided PCI reduced the rate of adverse CV events (13% vs. 22%, HR 0.56, 95% CI 0.38–0.83, *p* = 0.004).
COMPARE-ACUTE, 2017	NCT01399736	295 pts undergoing CR vs. 590 pts receiving C-OR *n* = 885	Index procedure or index admission (staged)	MACE: Death, non-fatal MI, revascularization, cerebrovascular events at 12 months FU.	MV FFR-guided PCI reduced the rate of adverse CV events (8% vs. 21%, HR 0.35, 95% CI 0.22–0.55, *p* < 0.001).
COMPLETE, 2019	NCT01740479	2016 pts undergoing CR vs. 2025 pts receiving C-OR *n* = 4041	Index admission (staged) or post-discharge	1. Composite of CV death and MI. 2. Composite of CV death, MI, and IDR at 36 months FU	1. MV PCI reduced the risk of CV death or MI (7.8% vs. 10.5%, HR 0.74, 95% CI 0.60–0.91, *p* = 0.004). 2. MV PCI reduced the risk of CV death, MI, and ischemia driven revascularization (8.9% vs. 16.7%, HR 0.51, 95% CI 0.53–0.61, *p* < 0.001).

C-OR, culprit-only revascularization; CR, complete revascularization; CV, cardiovascular; FFR, fractional flow reserve; FU, follow-up; HF, heart failure; IDR, ischemia-driven revascularization; MACE, major adverse cardiovascular events; MI, myocardial infarction; MV, multivessel; PCI, percutaneous coronary intervention; Pts, patients.

**Table 2 jcm-12-04550-t002:** Comparison between OCT and IVUS with specification of the characteristics and potential of each method.

Skills	OCT	IVUS
Assess left main lesion severity		++
Identify culprit lesion	++	+
Identify vulnerable plaque	+	
Evaluate calcium burden	++	+
Optimize stent implantation	++	+
Assess risk of distal embolization after PCI	++	
Evaulate stent failure	++	+
Reduce contrast		++

## Data Availability

Not applicable.

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
