# Peer review of "Moving toward Precision Medicine in Acute Coronary Syndromes: A Multimodal Assessment of Non-Culprit Lesions"

_jcm, 2023, doi:10.3390/jcm12134550_

Round 1
Reviewer 1 Report
I have read manuscript entitled "Moving toward precision medicine in acute coronary syndromes: multimodality assessment of non-culprit lesions" which is well written clinical review of current problem in invasive cardiology. The authors conducted an in-depth analysis of the current literature with a special focus on clinical trials. The authors noted the importance of plaque assessment.
strengths: - good summary of recent clinical trials with focusing on specific questions - nice operational algorithm which can be helpful for physician
weaknesses - the manuscript should be updated with documents published in 2023 - The authors should consider adding a table of the most important clinical trials with advantages and disadvantages of specific strategies.
The article is well written summary of actual knowledge about revascularization in multivessel coronary disease. Authors should update the references i.e.
Kaziród-Wolski K, Sielski J, GÄ…sior M, et al. Factors affecting short- and long-term survival of patients with acute coronary syndrome treated invasively using intravascular ultrasound and fractional flow reserve: Analysis of data from the Polish Registry of Acute Coronary Syndromes 2017-2020. Kardiol Pol. 2023; 81(3): 265–272.
Berntorp K, Rylance R, Yndigegn T, et al. Clinical Outcome of Revascularization Deferral With Instantaneous Wave-Free Ratio and Fractional Flow Reserve: A 5-Year Follow-Up Substudy From the iFR-SWEDEHEART Trial. J Am Heart Assoc. 2023;12(3):e028423. doi:10.1161/JAHA.122.028423
Yuki H, Kinoshita D, Suzuki K, et al. Layered plaque and plaque volume in patients with acute coronary syndromes. J Thromb Thrombolysis. 2023;55(3):432-438. doi:10.1007/s11239-023-02788-9
Some minor language issues i.e. verse 110 ,, in the coronary three”.
Author Response
We are thankful to the Editor and to the Reviewers for the thorough review and for the opportunity to submit a revised version of our manuscript entitled “Moving toward Precision Medicine in Acute Coronary Syndromes: Multimodality Assessment of Non-Culprit Lesions" (jcm-2466939).
In this revised version of the manuscript, we have addressed the insightful comments of the editorial committee and of the external reviewers. We believe that changes requested and made to the manuscript have significantly improved the quality of our paper.
All changes to the manuscript are highlighted in red within the text.
In this reply letter, each comment by the reviewers (bold) is followed by our response. For substantive changes made to the manuscript, we provide a description of what we did and where. Important sentences, paragraphs or sections in response to the comments have been included in italics.
REVIEWER #1
I have read manuscript entitled "Moving toward precision medicine in acute coronary syndromes: multimodality assessment of non-culprit lesions" which is well written clinical review of current problem in invasive cardiology. The authors conducted an in-depth analysis of the current literature with a special focus on clinical trials. The authors noted the importance of plaque assessment.
strengths: - good summary of recent clinical trials with focusing on specific questions - nice operational algorithm which can be helpful for physician
weaknesses - the manuscript should be updated with documents published in 2023 - The authors should consider adding a table of the most important clinical trials with advantages and disadvantages of specific strategies.
We thank the reviewer for his consideration regarding the manuscript.
In agreement with his observations and notes we have updated the Review with inherent documents published in 2023 on page 3, adding the appropriate references.
A dedicated table with most important clinical trials with their findings has been inserted (Table 1).
The article is well written summary of actual knowledge about revascularization in multivessel coronary disease. Authors should update the references i.e.
Kaziród-Wolski K, Sielski J, GÄ…sior M, et al. Factors affecting short- and long-term survival of patients with acute coronary syndrome treated invasively using intravascular ultrasound and fractional flow reserve: Analysis of data from the Polish Registry of Acute Coronary Syndromes 2017-2020. Kardiol Pol. 2023; 81(3): 265–272.
Berntorp K, Rylance R, Yndigegn T, et al. Clinical Outcome of Revascularization Deferral With Instantaneous Wave-Free Ratio and Fractional Flow Reserve: A 5-Year Follow-Up Substudy From the iFR-SWEDEHEART Trial. J Am Heart Assoc. 2023;12(3):e028423. doi:10.1161/JAHA.122.028423
Yuki H, Kinoshita D, Suzuki K, et al. Layered plaque and plaque volume in patients with acute coronary syndromes. J Thromb Thrombolysis. 2023;55(3):432-438. doi:10.1007/s11239-023-02788-9
References suggested by the reviewer have been incorporated into the text and bibliography

Reviewer 2 Report
I have read the work of Bellino et al about the bystander disease in ACS patients. Indeed data is mixed and debated. What is known is that non-culprit lesions should indeed be revascularized (aiming for a full revascularization) but WHICH ones and HOW is debated at the moment. The authors aimed through this narrative review to address this gap and connect the dots on what we know and what are the future perspectives in the field. The current ESC recommendation is IIb for revascularizing non-culprit and using FFR for such lesions. These recommendations are based mainly on 2 trials, the DANAMI and COMPARE ACUTE trials. The two trials, although looking at bystander disease in ACS, treated the lesions differently ( one staged culprit, one in the same ACS session).
The article reads well, it is more like an overview of the current literature. It parallels the two sides of the story, invasive physiology vs intracoronary imaging. Lately, the FLOWER-MI "shook" the results of DANAMI and COMPARE ACUTE and questioned the role of FFR in ACS. But even more recently, the FRAME MI study regained territory for FFR, showing that selective PCI of non-infarct-related artery lesions using FFR-guided decision-making was superior to a strategy of routine PCI based on angiographic diameter stenosis in patients with acute MI and multivessel disease. The story is not yet over.
We must acknowledge that the topic of this review is not new and several publications addressed it already (for example, one important work by Paradies et al in Eurointervention, DOI: 10.4244/EIJ-D-20-00957, and one similar work in the same JCM MDPI, again by an Italian team from Napoli, Piccolo et al, doi: 10.3390/jcm12072572). Nevertheless, I consider this worthy of publication because (1) the first work is already outdated and (2) the second work could be cited and (3) the authors would succeed to present their work in a novel way and not repeat the findings of Piccolo et al. I would have a few suggestions for this matter:
- improve the visuals of the review. Draw a graphical abstract with the timeline of the trials mentioned above, or even a study design figure that would include also the years of the studies. I will try to atach a model.
- you discussed the limitations of OCT & IVUS. Discuss also the limitations of FFR in ACS. Moreover, the use of FFR is still not fully adopted, nor used when requesting adjunctive functional assessment (pls cite: DOI: 10.1016/j.ijcard.2021.05.005).
- FFR is also debated in chronic coronary syndrome although the ESC recommendation in this setting is Ia (see FAME 3 trial DOI: 10.1056/NEJMoa2112299 and also this intriguing editorial https://doi.org/10.1093/ejcts/ezac036, you could cite them both and make a parallel).
- at the Vulnerable plaque section, add some OCT and NIRS suggestive intracoronary images of such plaques.
I am looking forward to seeing the revised version of the manuscript & I hope my comments are useful to the authors and will help bring this paper to a shape that could add more interest and value to the readership.

The English language and grammar need just minor revisions.
Author Response
We are thankful to the Editor and to the Reviewers for the thorough review and for the opportunity to submit a revised version of our manuscript entitled “Moving toward Precision Medicine in Acute Coronary Syndromes: Multimodality Assessment of Non-Culprit Lesions" (jcm-2466939).
In this revised version of the manuscript, we have addressed the insightful comments of the editorial committee and of the external reviewers. We believe that changes requested and made to the manuscript have significantly improved the quality of our paper.
All changes to the manuscript are highlighted in red within the text.
In this reply letter, each comment by the reviewers (bold) is followed by our response. For substantive changes made to the manuscript, we provide a description of what we did and where. Important sentences, paragraphs or sections in response to the comments have been included in italics.
REVIEWER #2
I have read the work of Bellino et al about the bystander disease in ACS patients. Indeed data is mixed and debated. What is known is that non-culprit lesions should indeed be revascularized (aiming for a full revascularization) but WHICH ones and HOW is debated at the moment. The authors aimed through this narrative review to address this gap and connect the dots on what we know and what are the future perspectives in the field. The current ESC recommendation is IIb for revascularizing non-culprit and using FFR for such lesions. These recommendations are based mainly on 2 trials, the DANAMI and COMPARE ACUTE trials. The two trials, although looking at bystander disease in ACS, treated the lesions differently ( one staged culprit, one in the same ACS session).
The article reads well, it is more like an overview of the current literature. It parallels the two sides of the story, invasive physiology vs intracoronary imaging. Lately, the FLOWER-MI "shook" the results of DANAMI and COMPARE ACUTE and questioned the role of FFR in ACS. But even more recently, the FRAME MI study regained territory for FFR, showing that selective PCI of non-infarct-related artery lesions using FFR-guided decision-making was superior to a strategy of routine PCI based on angiographic diameter stenosis in patients with acute MI and multivessel disease. The story is not yet over.
We must acknowledge that the topic of this review is not new and several publications addressed it already (for example, one important work by Paradies et al in Eurointervention, DOI: 10.4244/EIJ-D-20-00957, and one similar work in the same JCM MDPI, again by an Italian team from Napoli, Piccolo et al, doi: 10.3390/jcm12072572). Nevertheless, I consider this worthy of publication because (1) the first work is already outdated and (2) the second work could be cited and (3) the authors would succeed to present their work in a novel way and not repeat the findings of Piccolo et al. I would have a few suggestions for this matter:
We thank the reviewer for the judgment on our manuscript and its cultural value .
improve the visuals of the review. Draw a graphical abstract with the timeline of the trials mentioned above, or even a study design figure that would include also the years of the studies. I will try to atach a model.
We appreciate the reviewer’s comment. A dedicated figure with the management provided by the main studies regarding the issue has been produced. Moreover, new Table 1 summarizes the most relevant clinical trial comparing culprit-only vs. complete revascularization in patients with acute coronary syndrome.
you discussed the limitations of OCT & IVUS. Discuss also the limitations of FFR in ACS. Moreover, the use of FFR is still not fully adopted, nor used when requesting adjunctive functional assessment (pls cite: DOI: 10.1016/j.ijcard.2021.05.005).
We thank the reviewer for the interesting insight. At page 3 and 4 we discussed the limitations of the invasive functional assessment in acute coronary syndromes.
The rationale has been expanded and deepened as follows
We included the suggested reference in the bibliography:
“However, these findings cannot be automatically translated into the management of NCLs in patients with ACS. Compared to patients with SIHD, ACS patients have a different biological and clinical profile, that is reflected by the high risk of recurrent events. [30] In these patents, the NCL-related risk of adverse events may be closely related to plaque morphology rather than to functional significance. Therefore, the deferral of non-flow limiting stenoses may still expose to the risk of adverse events related to the presence of high-risk morphological features, such as a thin fibrous cap, high lipid burden and in-flammation. Moreover, transient changes in microvascular physiology in the acute phase of an MI may affect the reliability of both hyperemic and non-hyperemic indexes. [31] [32] [33] These changes occur not only in the territory of the infarct-related artery, but also in areas of myocardium supplied by NCLs, especially in case of large infarcts. [34] FFR is the whole cycle ratio between distal coronary pressure (Pd) and aortic pressure (Pa) in a condition of maximal hyperemia, usually induced by the administration of adenosine. However, in the acute or subacute phase of an MI, hyperemic flow may be reduced and tends to normalize within months from the acute event. [35] Several alterations in mi-crovascular function are responsible for the reduction of hyperemic flow, such as a reduced response to adenosine, the enhanced microvascular vasoconstriction, and the microvas-cular compression due to edema and increased end-diastolic pressures. [36] Because the reliability of FFR measurements is related to the induction of the maximal hyperemic state, a reduced hyperemic flow may induce the underestimation of the functional sig-nificance of NCLs. [37] [38] Furthermore, several real-world data showed that FFR is significantly underused by the interventional cardiology community in the assessment of angiographically intermediate stenoses, even in patients with stable CAD. [39] Considering the acute clinical setting, the additional procedural time, and the need for adenosine administration, FFR might be even less adopted in patients with ACS.
Moreover, additional questions about the role of FFR in patients with MVD were raised by the FAME 3 study. In this trial, an FFR-guided PCI strategy did not meet the noninferiority margin compared to coronary artery bypass grafting in patients with three-vessel disease. [40] However, although about 40% of the patients enrolled had a non-ST segment elevation ACS, this trial was not specifically designed to assess the role of FFR in patients with ACS and MVD, and STEMI was an exclusion criteria.
On the other hand, NHPRs measure resting Pd/Pa during the entire cardiac cycle or during specific phases of the diastole, depending on the specific index. [38] However, during the acute phase of an MI resting flow may be increased, probably due to the compensatory hyperkinesia in non-infarct related territories. [34] [36] Therefore, NHPRs may overestimate the functional significance of NCLs.”
FFR is also debated in chronic coronary syndrome although the ESC recommendation in this setting is Ia (see FAME 3 trial DOI: 10.1056/NEJMoa2112299 and also this intriguing editorial https://doi.org/10.1093/ejcts/ezac036, you could cite them both and make a parallel).
According to the reviewer suggestion, at page 4, we discussed the results of the FAME 3 study about the role of FFR-guided PCI in patients with multivessel disease:
Moreover, additional questions about the role of FFR in patients with MVD were raised by the FAME 3 study. In this trial, an FFR-guided PCI strategy did not meet the noninferiority margin compared to coronary artery bypass grafting in patients with three-vessel disease. [40] However, although about 40% of the patients enrolled had a non-ST segment elevation ACS, this trial was not specifically designed to assess the role of FFR in patients with ACS and MVD, and STEMI was an exclusion criteria.
7) - at the Vulnerable plaque section, add some OCT and NIRS suggestive intracoronary images of such plaques.
Reviewer's commentary enhances the iconography of the manuscript.
According to it, OCT images have been added with appropriate captions to complement them.
Unfortunately, none of the Centers at which the authors work are able to provide NIRS images.
I am looking forward to seeing the revised version of the manuscript & I hope my comments are useful to the authors and will help bring this paper to a shape that could add more interest and value to the readership.

Round 2
Reviewer 2 Report
The authors only partially revised this paper. They state that the suggested discussions were now included and the references were included as well but I don't see any of the references I have suggested ( DOI: 10.1016/j.ijcard.2021.05.005 and https://doi.org/10.1093/ejcts/ezac036). I am looking forward for a better and thorough revised version.
No major issues.
Author Response
We apologize to the reviewer for the error made.
The suggested references on coronary physiology have been inserted on page 6 and in the corresponding position in the bibliography.